# Towards Community Rooted Research and Praxis: Reflections on the BSS Safety and Youth Justice Project

Uriel Serrano [1,*], David C. Turner III [2] , Gabriel Regalado [3] and Alejandro Banuelos [4]

1    Sociology and Critical Race and Ethnic Studies, University of California, Santa Cruz, CA 95064, USA
2    Ralph J. Bunche Center for African American Studies, University of California, Los Angeles, CA 90095, USA; dcturner@ucla.edu
3    Center for the Study of Slavery and Justice, Brown University, Providence, RI 02906, USA; gregalado@sjli.org
4    Brothers, Sons, Selves, Los Angeles, CA 90023, USA; banuelos.alej@gmail.com
*    Correspondence: serrano@ucsc.edu

**Abstract:** This article focuses on the Brothers, Sons, Selves (BSS) Safety and Youth Justice project to describe what we refer to as a *Community Rooted and Research Praxis* (CRRP) approach. BSS is an organizing coalition for boys, young men, and masculine-identifying youth of color that works to decriminalize communities of color. In 2018, BSS developed a survey to capture how safety and justice is experienced by youth of color across multiple contexts and institutions in Los Angeles County. With over 3000 surveys collected, the findings have now been used to promote racial equity and decriminalize youth at the local and state level. Building on a Black Radical Tradition, including abolitionists struggles against the carceral state, in this paper, we name *CRRP* as a framework to describe BSS's community engaged scholarship. In other words, we contend that the CRRP approach is a mode of community engaged scholarship that brings together youth, university affiliated adults, and community organizations to engage in youth participatory action, research, political education, and collective struggle.

**Keywords:** community-based research; youth participatory action research; youth organizing; youth activism; Black Radical Tradition; abolition

## 1. Introduction

> "I believe that when we are in relationship with each other, we influence each other. What matters to me, as the unit of interest, is relationships".—Mariame Kaba, *We Do This 'Til We Free Us*

During the fall of 2018, over 30 Brothers, Sons, Selves (BSS) youth and youth workers gathered on a Saturday morning in South Central Los Angeles for a workshop on using research and data for social change.[1] BSS is an organizing coalition for boys and young men of color that works to decriminalize communities of color. The workshop was led by two of the co-authors, and it introduced youth to quantitative and qualitative research approaches. The workshop also explored the limitations and possibilities of research, including how social science research has been used to harm and pathologize Black communities and other communities of color (Hartman 2019; Ferguson 2004). At the end of the workshop, youth asked if young people themselves were included in developing and implementing studies on them. Realizing that youth of color are rarely included in this process (Jacquez et al. 2013), they decided to embark on the creation of a youth participatory action research (YPAR) project: The BSS Safety and Youth Justice Project. This project, which we describe later, has been used to inform campaign efforts and movement building.

A new generation of social actors has emerged in the ongoing movement against the harms of racial capitalism and anti-Black state violence (Johnson and Lubin 2017). Building on prior movements, Black, Latinx, and other youth of color are meeting police violence

and repression with protest and a wide range of oppositional strategies (Turner 2021; Terriquez and Milkman 2021; Serrano 2020; Escudero 2020; Ransby 2018). For example, in Los Angeles County, youth activists successfully advocated for the defunding of the Los Angeles School Police Department by $25 million in 2020 (Tat 2020). Besides youth-led protests and rallies, and the sharing of testimonies during Los Angeles Unified School District (LAUSD) school board meetings, community engaged research, including the BSS Safety and Youth Justice Project, made up the "wide range of oppositional strategies" by youth activists.

At its most basic, community engaged research projects—also referred to as community engaged scholarship, participatory action research, youth participatory action research, etc.—are collaborative processes between a community and researcher(s) (Torre 2014; Anyon et al. 2018). By involving community members at multiple levels and incorporating and validating various sources of knowledge, researchers have argued that a central goal of this method is to understand and support a specific community's needs and foster social change accordingly (Merenstein 2015). Youth participatory action research (YPAR) emerged as a bridge to engage young people in this process. YPAR involves partnerships between youth and adults to research and co-construct knowledge on youth lives and their communities (Cammarota and Fine 2010). The information that comes out of YPAR projects is meant to address social problems and center youth as experts (Anyon et al. 2018). Yet, Gordon da Cruz's (2017) review of community engaged research literature finds that social justice, or addressing inequality, is not always at the forefront of university-led community engaged research projects. Gordon da Cruz (2017) proposes *critical community engaged scholarship* (critical CES) as a corrective.

Critical CES, which we elaborate on in the section to follow, proposes that critical race theory can improve the effectiveness of community-engaged research with social justice aims (Gordon da Cruz 2017, p. 368). To this end, Gordon da Cruz proposes four questions for "community–university research collaborations aiming to dismantle systemic sources of injustice" (p. 365): *Are we collaboratively developing critically conscious knowledge?*; *Are we authentically locating expertise?*; *Is our work grounded in asset-based understandings of community?*; and *Are we conducting race-conscious (instead of color-blind) research and scholarship?* While critical CES does not explicitly engage YPAR and youth organizing, we find the four questions generative and engage with them as an organizing tool for this paper.

In this paper, we draw on the BSS Safety and Youth Justice Project and center BSS youth and insights from their collective struggles against the carceral state to engage and reflect with these questions as both a building on and a departure from critical CES. To do so, we also draw on insights from the Black Radical Tradition and youth organizing, including abolitionist analyses of the carceral state (Davis et al. 2022; Kaba 2021; Berger 2014; Felker-Kantor 2018). In so doing, this approach departs from critical CES in that it is not guided by notions of reforming the criminal legal, and economic systems that generate the harms experienced by community research participants. It is guided by the imperative to fundamentally disrupt and radically transform them. As Dylan Rodriguez observes, "reform movements tend to simultaneously obscure and reproduce normalized conditions of terror" (Rodríguez 2021, pp. 156–57). In contrast, the Black radical and abolitionist approaches seek to comprehensively dismantle the institutions and systems that perpetuate those conditions of terror.

Cedric Robinson ([1983] 2021) described the Black Radical Tradition as "a collective consciousness informed by the historical struggles for liberation and motivated by the shared sense of obligation to preserve the collective being, the ontological totality" (p. 171). Collective struggle thus becomes the basis for consciousness, knowledge, and addressing the contradictions and harms of racial capitalism, including the violence of the criminal legal system and the carceral state (Battle and Serrano, forthcoming). As Robinson ([1983] 2021) writes in *Black Marxism*, the developments of racism and capitalism are mutually constituted. As such, he contends that racial capitalism created the modern world via anti-Black chattel slavery and colonialism. Additionally, as community organizers and scholars now contend,

the carceral state is also part of the social fabric of the United States (Hernández 2017; Gilmore 2007). In this sense, we understand the carceral state as formal institutions *and* punitive logics that produce "sanctioned or extralegal production and exploitation of group differentiated vulnerability to premature death" (Gilmore 2007, p. 261). It is within this context that we name *community rooted research and praxis (CRRP)* as a framework to describe BSS's YPAR approach.[2]

### 1.1. Participatory Action Research

Perhaps most aligned with our CRRP framework is what is commonly referred to as participatory action research, or 'PAR.' PAR is an amalgamation of practices that seek to *include* the community in the knowledge-production process about a problem or issue that is relevant to that community (Reason and Bradbury 2008). The notion of "participatory" brings with it the idea that the community can and should be a part of the research process, from the creation of a research question, identifying key literature that exists on the topic, creating research instruments and collecting data, analyzing data, to reporting findings (Reason and Bradbury 2008). By involving the community in this process, a scholar has worked with the "experts" on a local issue since this topic of research is typically pertinent to their everyday lives and experience.

The "action" portion of PAR stems from human and civil rights activists all over the globe who engage in activities that attempt to alter oppressive conditions (Reason and Bradbury 2008). By incorporating the "action" element in PAR, it is essential to develop a practical use for the research being conducted by the community. Some scholars have used PAR to change and/or create local policies and address inequities in local and civic institutions; they have used it as tools for academic intervention, and have used PAR to increase the literacy and civic participation of national and global citizens. PAR has the flexibility to adapt research methodologies to the needs of the communities. For example, suppose a community wanted to analyze the effects of neoliberal education on their academic achievement (see Scorza 2013). In that case, they can incorporate focus groups, interviews, and surveys into their analysis. By giving the community tools to conduct the research and by engaging in the research process with them, the community becomes empowered and begins to break barriers between the researcher and the community (Smith 1999).

As outlined by Wicks et al. (2008), at the foundation of PAR are the following three pieces of research identity: the personal, the political, and the philosophical. By engaging with other scholars who facilitate participatory modes of inquiry, Wicks, Reason, and Bradbury were able to identify "experiences" as the common trend joining PAR researchers and community members. When the researcher owns their identity and social location, the notion of "objectivity" loses relevance given that their project might now include the lives, epistemologies, and ideas of the communities in which one works with. Thus, PAR argues that the people most affected by social issues are best equipped to solve them, seeing that they know them and live with them every day.

That that end, PAR practitioners draw from different schools of intellectual influence, ranging from critical pedagogy and conscientious liberation as defined by Paulo Freire (1970), social capital as defined by Pierre Bourdieu (1984), critical race theory as defined by Delgado and Stefancic (2017), and Black feminist theories as defined by Patricia Hill Collins (1986) and Deborah King (1988). For example, Torre et al. (2012) draw on the work of Paulo Freire and critical theory to outline a critical PAR approach. Informed by a project designed to examine how youth experience inequality across contexts, Torre and colleagues argue that a critical PAR approach "shifts the gaze from 'what's wrong with that person?' to "what are the policies, institutions, and social arrangements that help form and deform, enrich and limit, human development?" (p. 179). Importantly, critical PAR is grounded in various forms of knowledge, expertise, and methodological approaches to refuse "the distinction between theoretical and applied, and science and advocacy" (Torre et al. 2012, p. 182). With such theoretical routes, PAR finds itself in a unique space to interrogate how systems of oppression converge, and apply a unique perspective to social

phenomena based on the social location and understandings of the participants. It is both a pedagogy and a methodology, making it a transformative way to view research and use design research for justice.

PAR can also be a transformative process for adult community members and youth. As a critical pedagogy and methodological tool with which to conduct research with youth, scholars from various disciplines utilize YPAR to engage youth, ranging from public health issues to school inequities. YPAR, in the same way as the critical PAR described by Torre et al. (2012), is conducted alongside youth as opposed to on youth, which places youth and their needs at the center of each study (Duncan-Andrade and Morrell 2008). If a facilitator works with youth, they potentially change school policies and culture and even save lives. As such, YPAR projects have also been facilitated and developed in youth organizations. Thus, the development of YPAR projects is not limited to university and community partnerships. Given this context, our CRRP approach blurs the lines between community and the university to consider how some of us are first and foremost grounded in our communities while *in* the university but not *of* it.

### 1.2. Critical Community Engaged Scholarship (Critical CES)

To move us closer to an approach informed by social justice that more effectively addresses structural harms, critical CES proposes that community-engaged scholarship informed by critical race theory—"which interweaves strategies, processes, and worldviews with emancipatory aims into publicly engaged work"—could improve the effectiveness of community-engaged approaches, particularly as it pertains to social justice (Gordon da Cruz 2017, p. 368). For academic institutions embracing community-engaged research, Gordon de Cruz proposes that explicitly naming social justice as a goal has the potential to disrupt how dominant cultural structures, ideologies, and practices negatively impact minority communities. In other words, "a professed commitment to the "public good" might not be enough to forefront questions about how community-university research collaborations and knowledge production can influence actual policies, laws, and/or cultural practices that impact the lives of nondominant community members". (Gordon da Cruz 2017, p. 365).

As Gordon da Cruz (2017) proposes, four critical questions should be asked in university and community collaborations: Are we collaboratively developing critically conscious knowledge?; Are we authentically locating expertise?; Is our work grounded in asset-based understandings of community?; and Are we conducting race-conscious (instead of color-blind) research and scholarship? This approach and the four questions proposed are generative for university partnerships, but as we demonstrate, this approach does not quite capture how BSS has been using YPAR projects and how we as university-affiliated researchers have supported this work. Furthermore, our CRRP approach describes how youth are fully integrated, engaged, and leading YPAR projects in their efforts to decriminalize their communities and forge abolitionist alternatives. As such, we argue that the prevailing literature on community-engaged scholarship has yet to recognize or even consider the vitality of abolitionist definitions of justice.

For instance, Gordon da Cruz's (2017) concept of a critical-theory-based and justice-focused engagement in CES stresses the need to "explicitly name and define justice", but does not identify precisely what is generating the harmful outcomes that necessitate critical and justice-focused approaches. She references the societal conditions in which so-called "minoritized" and historically marginalized communities are systematically harmed, but does not explicitly mention the interlocking forces of policing and prison systems, heteropatriarchy, and racial capitalism ubiquitously embedded in institutions, including academia, that purvey that harm. Therefore, her critique of the prevailing discourse on the "public good" in CES also lacks a cogent meaning of what justice actually entails. As we discuss below, our approach to CRRP is vigilantly conscious of how these interlocking forces are not only ineffective in addressing harm, but are the very conduits through which

much of the harm in our communities is generated and therefore must rigorously and immediately be confronted, critiqued, and challenged towards eventual abolition.

Gordon da Cruz's concept of critical CES was conceived prior to the 2020 BLM uprisings[3] that brought the language of abolition into the fold of mainstream political discourse on public safety, but as Robin D. G. Kelley (2021) reminds us, "abolition is neither new nor hopelessly utopian". Regardless of whether or not it is explicitly named, the abolitionist perspective—described by Mariame Kaba (2021) as "a political vision, a structural analysis of oppression, and a practical organizing strategy" to rid society of policing and prison systems rooted in heteropatriarchy and racial capitalism—has remained an ever present commentary in political critique throughout the last quarter of the 20th century and into the 21st century. The omission of this perspective in community-engaged scholarship severely limits Gordon da Cruz's justice-focused aspirations of the transformative substance needed to effectively operationalize critical CES.

For instance, Gordon da Cruz seeks to hold universities accountable in upholding "their democratic commitments" but does not account for Angela Y. Davis (2005) analysis of how democracy in general is pervasively denied by policing and prison systems that uphold the enduring legacies of slavery and colonialism. Gordon da Cruz's concept of justice is also built on "welfare economic" theories in lieu of an explicitly anti-capitalist framework that recognizes how racism and capitalism are mutually constituted (Robinson [1983] 2021) as well as how heteropatriarchy—a social system in which patriarchy and heterosexuality are normalized—and capitalism are mutually constituted (Taylor 2017; Arvin et al. 2013). Critical CES can therefore be interpreted as a *reformist* neoliberal conjecture instead of a justice-focused and community-based approach towards abolition.

For example, Gordon da Cruz (2017) offers two different hypothetical vignettes to exemplify her Critical CES approach and differentiate it from community engaged scholarship. Both vignettes introduce the imaginary Professor Jones, a white faculty member in a criminal justice department interested in examining policing and contributing to the "public good". In the second vignette, where Professor Jones is more driven by community interest and an approach more closely aligned with Critical CES, the proposed project that is positioned as the lead reform for addressing police violence is a research project on police training and implicit bias (2017, pp. 376–78). In spite of community feedback within the hypothetical example that situates the problem in a structural analysis that implicates police overfunding compared to other public and social services, he ultimately lands on the liberal reform of increased police training. In abolitionist guides to transforming the carceral state, such as the 8ToAbolition framework[4], advocating for increased police training is discouraged, as increased training gives more money to law enforcement. Instead, alternatives to policing and the carceral state are encouraged by abolitionists, specifically the divestment of funds from policing and instead investing in communities (Ransby 2018). For research to be truly community engaged, it must support the community in transforming relationships of power, not offer the state additional tools and mechanisms to increase state power. We suggest that Gordon da Cruz's framework for Critical Community Engaged Scholarship must be extended.

## 2. Towards a Community Rooted Research and Praxis Approach

### 2.1. The Brothers, Sons, Selves Coalition and the Safety and Youth Justice Project

BSS is a Black, Latinx, and Southeast Asian coalition developing the leadership of boys, men, and masculine-identifying folks of color to address conditions of criminalization enveloping our communities and schools. BSS is made up of young people, organizers, and youth workers from seven community-based organizations across Los Angeles County, including Innercity Struggle, East LA Weingart YMCA, Community Coalition, Brotherhood Crusade, Youth Justice Coalition, Social Justice Learning Institute, and Khmer Girls in Action. All seven partner organizations are in one way or the other anchored in the realities and experiences of Black and Brown people in Los Angeles (Turner 2021; Serrano and Terriquez 2021).

Since 2011, BSS has shaped how public policy impacts Black and Brown youth by refining the organizing skills of youth of color to meaningfully lead campaigns to abolish school-based carceral approaches and advocate for structures of care and healing in their place. BSS meets weekly to strategize, share consciousness raising knowledge, critique the carceral state, and learn to embrace and love one another through struggle. What keeps BSS together is an enduring belief in Black and Brown youth and their visions for the future. Serving as a testament of their leadership development, BSS staff is partly made up of former youth leaders and local community members.

For the past year, BSS has been setting the foundation for a campaign called the Los Angeles County Youth Bill of Rights (YBoR). YBoR is a policy vision for Los Angeles County youth anchored in these three campaign pillars: Decriminalization Now, Fund Youth Futures, and The People's Education. Each campaign pillar holds policy demands youth want to see enacted in LA County. For example, in Decriminalization Now, the call is to raise the minimum for youth arrest to 16 years old with a large vision of ending punitive practice completely. Another demand includes the investment and expansion of existing initiatives creating an infrastructure to replace youth criminalization. Fund Youth Futures seeks to pay young people to prepare them for different careers and expand access to quality mental health in their communities. Finally, The People's Education calls on LA County to create an incentive fund to support the creation of local youth decision-making bodies and to enforce a California state mandate requiring schools to have one counselor for every 300 students. BSS youth leaders created these campaign pillars after analyzing survey data they collected in 2018.

In 2018, BSS developed a survey to capture how safety and justice are experienced by youth of color across multiple contexts and institutions. BSS youth spent hours deliberating the research questions, survey design, data collection protocols, analysis, and data dissemination. BSS partner organizers and BSS staff members worked to facilitate workshops titled, "research 101" and "using research for social change" to build the preliminary BSS survey. The survey questions were finalized at the end of 2018, and the survey was administered from January to June of 2019 by our partner organizations, their youth, and a youth civic engagement team. A total of 3733 surveys were administered, and 3378 surveys made it into the final analysis. The themes covered in the survey ranged from perceptions of law enforcement, to how youth experience criminalization, policing, and incarceration in Los Angeles County, including their schools.

After we received the final data set, BSS youth workers developed training for youth leaders to discuss data analysis, data visualization, and survey findings. Youth leaders analyzed the BSS data through a series of workshops where they collaborated in small teams facilitated by BSS staff to understand and create data statements and visualize the findings; sit and ponder with different data points; identify disparities across region, race, gender, sexuality, and involvement with the justice system; and connect data points to their personal stories. This deep engagement with the survey data only confirmed what BSS youth already knew and suspected, that youth of color in Los Angeles County are targeted by police, criminalized, and harmed across institutions. Rejecting criminalization and inequality as their only social reality, the survey protocol also asked youth and other community members about their aspirations for the local community regarding safety, education, and racial, gender, and economic justice. For example, respondents were asked to express their views and desires around school safety, the presence of police in schools, and funding for mental health resources and other academic and social programming.

For BSS, data provide a tool to indict systems of oppression targeting boys and men of color, specifically, and all youth of color more broadly. For example, the Los Angeles County Youth Bill of Rights was the first project BSS youth leaders undertook using the BSS Safety and Youth Justice survey findings. The Youth Bill of Rights is an original document created by BSS young people articulating their vision for youth justice in Los Angeles County. It is anchored in three campaign pillars (Decriminalization Now, Fund Youth Futures, and the People's Education) with demands connected to the themes of each campaign pillar. These

three campaign pillars emerged from months of data analysis and discussions connected to the BSS Safety and Youth Justice Survey. Involved in this process were BSS youth leaders, youth workers, and other organizational leadership. Drawing from the survey findings and their own lived experiences, BSS youth leaders used the three campaign pillars and their advocacy to articulate the need to reduce youth contact with the carceral state, invest in the livelihood of young people and their positive health outcomes, and transform education in LA County.

### 2.2. BSS and CRRP

Our CRRP approach is a mode of community-engaged scholarship that brings together youth, university affiliated adults, and community organizations to engage in YPAR projects, political education, and collective struggle. This approach is not driven by a desire to advance the professed commitment to "public issues" or the "public good" by universities but instead to seriously consider how our relationship with the university is a "criminal one" (Moten and Harney 2013). As Moten and Harney (2013) argue:

> "it cannot be denied that the university is a place of refuge, and it cannot be accepted that the university is a place of enlightenment. In the face of these conditions one can only sneak into the university and steal what one can. To abuse its hospitality, to spite its mission, to join its refugee colony, its gypsy encampment, to be in but not of—this is the path of the subversive intellectual in the modern university". (p. 101).

In other words, we highlight how we use university resources and the skills and tools we have gained in the name of collective struggle. Particularly, collective struggle with youth in Los Angeles County who are currently pushing against the carceral state in their schools and neighborhoods. At its core, CRRP, as we propose, is a mode of scholarship and inquiry that inverts the power dynamic of who is responsible for creating knowledge and the purposes for creating knowledge. CRRP is not a project interested in creating knowledge for knowledge's sake, nor is CRRP interested in merely partnering with community members to study conditions collectively. CRRP is fundamentally about using university tools and training to build power with communities impacted by systemic oppression under the leadership and direction of the community. It positions researchers not only as thought partners but as co-conspirators working under the guidance of the community to transform our collective conditions.

As the following sections will demonstrate, we reflect on BSS's YPAR project to chart a CRRP approach (see Figure 1). In these reflections, we do not shy away from discussing how our social location within BSS informs CRRP. Specifically, our investment and commitment to improving youth's material conditions and well-being in Los Angeles County are explored. As such, we want to acknowledge that we have all played various roles in BSS. Uriel, currently a PhD candidate, has supported the BSS Action Research Committee and respective YPAR projects for the past four years. While completing a doctoral program, David was the BSS manager from 2018 to 2022 and completed his PhD in 2021. Gabriel, also a PhD candidate during this time, is currently a Youth Justice Organizer with the Social Justice Learning Institute's Policy and Advocacy Division. In 2019, Alejandro joined BSS as the Youth Campaign Coordinator. Having arrived after data collection, Alejandro has worked alongside youth leaders to make sense of the BSS survey data by paying attention to how race, gender, sexuality, class, geographical context, and experiences with the prison industrial complex shape the experiences of youth. It is from these vantage points that we offer the reflections that follow.

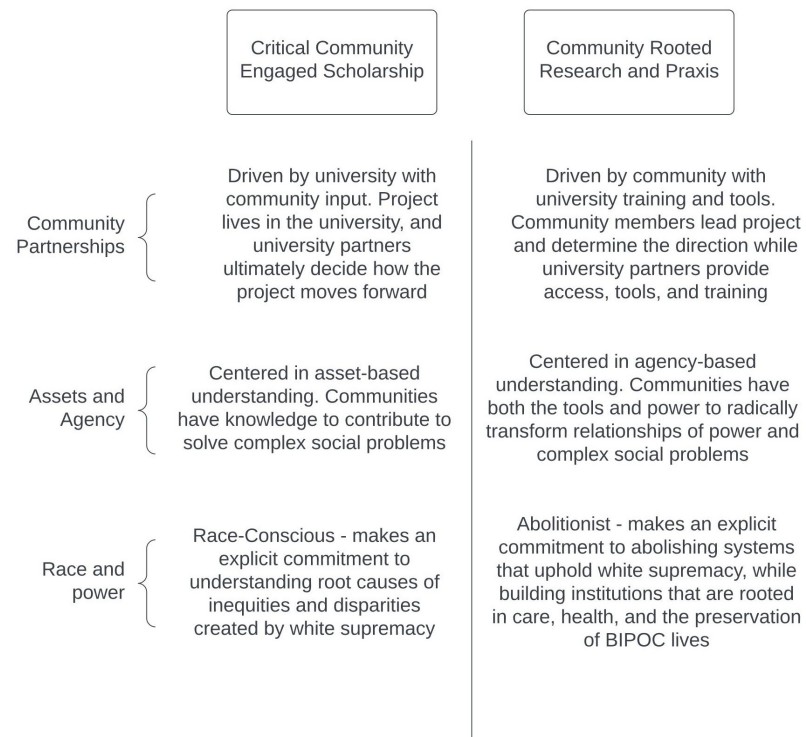

**Figure 1.** Difference between CCES and CRRP.

### 2.3. Reflections on Critically Conscious Knowledge and Locating Expertise

Introduced by Freire (1970), critical consciousness is a pedagogical approach rooted in an awareness of systemic inequality that leads to a practice rooted in social transformation and resistance. Informed by critical race and LatCrit theorists, Gordon da Cruz (2017) proposes that "critically conscious knowledge acknowledges and is critical of how race and racism and the intersection of racism with other forms of subordination influences people's experiences and opportunities for success" (p. 372). Critically conscious knowledge also recognizes the role of cultural assumptions and racialized deficit perspectives in the production of inequality. Moreover, Gordon da Cruz (2017) warns us that not all projects lead to the development of critically conscious knowledge, which is especially true when university affiliated researchers fail to interrogate how their "own social location could influence assumptions about how societal power structures or dominant cultural assumptions create and sustain structural inequity" (p. 372). Thus, the aim of critical CES is to collaboratively develop critically conscious knowledge in community and university partnerships.

Gordon da Cruz (2017) also encourages researchers to "gather expertise on the causes and impacts of racialized and racist cultural practices" (p. 373) to develop critical consciousness knowledge. This is carried out with the goal of collaborating with community members and privilege the expertise of those most impacted by inequality (Gordon da Cruz 2017; Ladson-Billings and Tate 1995; Solorzano and Bernal 2001). At the university level, Greenberg et al. (2020) describe a similar process where undergraduate students are valued for the expertise they bring in researching social problems in their communities. Their approach signals the importance of moving from "community-engaged" to "community-driven" processes. Here, we build on Greenberg et al.'s (2020) call to move beyond "community engaged" to reflect on questions around critical consciousness and expertise because the BSS Safety and Youth Justice Project was initiated and driven by the existing critically conscious knowledge of BSS youth.

As you might recall from the introduction, it was the interrogation of how research has played a role in pathologizing youth of color, their neighborhoods, and their absence from the development of studies on them that sparked an interest in a YPAR project. In this instance, youth critical consciousness, or critically conscious knowledge, was not something

that required "development", "locating", or "gathering". Instead, it was built through ongoing political education, and it drove the creation of a youth-led and youth-informed YPAR project, including the research questions guiding the project. This is key because critical consciousness is often positioned only as a developmental outcome for youth of color. Moreover, it inverts community and university partnerships from one of "gathering expertise" to one where the community *uses* the university and its resources.

Our *community rooted research and praxis* approach embraces the consciousness and knowledge already existent in communities that is developed through collective struggle and by bearing the brunt of racial capitalism. This embracement is informed by our approach of being *in community* as opposed to partnering with community. Not only do we come from similar backgrounds as BSS youth, but at times, their experiences are a reflection of our own experiences. As individuals affiliated with universities, however, we leverage our resources and access to grants, data analysis software, and existing networks to be *in community* with BSS youth. We define being "in community" as having a material stake in turning research into legitimate campaigns and struggles. For the BSS Safety and Youth Justice Project, that included letting youth decide what research questions guided data collection, survey design, and how the findings were used.

Our role in that process included providing guidance and feedback when necessary and introducing them to methodological approaches appropriate for their questions. As such, our CRRP approach includes being *in community* by collaborating on workshops on research methods, including data collection, analysis, and dissemination. These workshops were developed with the goal of allowing BSS youth to make their own decisions about the development of the BSS Safety and Youth Justice project. When appropriate, we also interrogated how research has been used to pathologize communities of color, especially Black neighborhoods. As such, workshops also explored how research has been used in campaign efforts by partner organizations and youth groups in other states. During and after workshops, the young people wrote the research questions, decided to design a survey, and wrote the survey questions after learning about survey methodology. In other words, we were not partnering with the local community to develop projects that centered our own research interests. Instead, we provided the tools, software, funding, and other infrastructure necessary for BSS youth to develop their project. Thus, to be *in community* is to occupy a fugitive space in the university—it means that we have stolen university knowledge, resources, and capital and have redistributed it to the community (Moten and Harney 2013). After data collection and analysis, we continued this process by supporting youth in using the survey in their campaign efforts to decriminalize youth. To date, the findings from this survey have helped shift state policy on school discipline and informed the allocation of $25 million towards Black Student Achievement in the Los Angeles Unified School District.

In this fugitive space, we blur the lines of university and community partnerships by using our access to resources to support BSS youth, their projects, and their visions of social transformation. More importantly, we position critical conscious knowledge and "expertise" as rooted in collective struggle. As Robin Kelley writes about the Black Radical Tradition, "progressive, social movements do not simply produce statistics and narratives of oppression; rather, the best ones do what great poetry always does: transport us to another place, compel us to relive horrors and more importantly, enables us to imagine a new society" (Kelley 2002, p. 9). This approach also compels us to recognize that young people can transform their material conditions.

### 2.4. Reflections on Asset-Based Understandings

Gordon da Cruz (2017) asserts that asset-based understandings of community are inherently anti-deficit—meaning that there should always be an analysis of power and oppression. Furthermore, researchers should not approach communities as being the problem in their own issues. Undergirding her analysis of researchers moving towards Critical CES is an asset-based understanding of that community—meaning that communities have much

more to offer and are not inherently "lacking" both social and cultural capital (see Yosso 2005). For example, in her second vignette with Professor Jones that is more favorable for Critical CES, Professor Jones asks the community about the kind of studies would support the community the most. In this example, community members suggested studies that led to structural and power-based analysis, departing from the "blame the victim" narratives that supported the first vignette. In the second vignette, the community members were seen as assets as opposed to accessories of a study, and their insight was taken and used to ultimately suggest a study about implicit bias and training.

While this is an example of using asset-based approaches in community research, we argue that asset-based understandings of the communities we work with do not go far enough. For CRRP, asset-based approaches to communities are assumed. Instead, we assert that CRRP is *agency-based*, meaning that community-rooted research and praxis fundamentally believe that communities have the inherent power and ability to shift their material conditions and change relationships of power (see Turner 2021). An agency-based understanding of communities is rooted in grassroots organizing as a vehicle to cultivate the collective human agency of minoritized groups, specifically to shift policy, practice, culture, and society more broadly (Terriquez 2015). For example, when engaging with the the BSS Safety and Youth Justice Survey, we already believed that the young people had assets. For the authors of this manuscript, we walked the same streets they did—we attended the same schools, caught the same buses, and lived in the same communities. The community's assets were our neighbors, our families, and ourselves. The survey was a tool used by our youth to assert youth power. When BSS youth leaders such as Jonathan Calles used BSS survey data to advocate for the elimination of willful defiance school suspensions for grades K-8, or when youth leaders such as Christian Wimberly or Emmanuel Karunwi used BSS survey data to advocate for the defunding of school police, the survey was used as a strategy to build political power. It was not simply a partnership with the community; the community *led* the project with adult support, which is fundamentally different from simply taking their input on a study design for a project that benefits the community. It is a project that the community designed and for which they had access to university tools, training, and infrastructure to execute.

For us, creating space for young people to assert their power and agency means getting them the tools to facilitate the type of change they want to see and move out of the way. Often in youth organizing, adults can play a role in harming the leadership and agency of young people. In fact, some adults have worked to co-opt youth-led movements and organizing in ways that fail to challenge state power or subvert the integrity of youth demands (see Clay and Turner 2021; Kwon 2013; Tuck and Yang 2013; Turner 2020). We argue that we must go further than having an asset based analysis not because we do not believe an asset-based analysis is important. We assert the need for an agency-based analysis and orientation so that communities retain their power to challenge—meaning they can say no, they can push back, and they can shift the direction of any project, campaign, or initiative as they see fit in the moment. In 2019, the youth leaders of the policy committee for the BSS voted unanimously to support Assembly Bill 392—the California Act to Save Lives. This bill updated California's use of force policy for law enforcement. Led by our partners at the Youth Justice Coalition, BSS youth leaders and partners would participate in phone banks, in-district delegation visits, and other key campaign activities to assist the Youth Justice Coalition and other co-sponsors of the bill, which was signed into law in the fall of 2019. It was the only policy presented to the youth at the time that had unanimous support among other policy priorities, and the youth were adamant about supporting it. As Clay and Turner (2021) suggest in their recommendations for practitioners, one key thing that adults can do is "encourage and amplify the voices of youth and community members in ways that go beyond typical state-sanctioned bureaucratic solutions (p. 30)". By investing in youth agency, we empower youth to move beyond simple "youth police dialogues" or technical solutions such as body cameras that expand police power to more actionable solutions such as taking police power and their abilities to enact violence away.

The BSS survey that the young people collected affirmed their strategic shift to support this initiative, where 1 out of every 20 youth who took the survey reported being physically harmed by law enforcement in some capacity and 39% of all youth who were harmed by law enforcement had never been arrested or officially detained by law enforcement.

Being driven by agency means seeing your youth as facilitators of their liberation. Take, for example, the campaign to defund school police in LAUSD. The youth from our partner organizations made a unanimous decision to join current efforts to defund the police in the wake of the George Floyd/Breonna Taylor protests, and their decision led to our coalition's deep involvement in the campaign. Youth such as Zae Ortiz, an Afro-Latinx young man, spoke about his experiences with school police in the neighboring Compton Unified School District. He highlighted how CUSD police had him in custody for nearly nine hours before informing his parents, and that he and his peers were targeted for no apparent reason. He connected what happened to him in Compton Unified to what happens to students daily in LA Unified, and he used BSS survey data and published university reports to accomplish it. By positioning youth such as Zae as facilitators of their own liberation, by investing in their agency, they are able to lead other youth, adults, and their communities closer to justice. While asset-based approaches to community engaged research are important, we assert that we must go further, we must believe that young people and communities can change their conditions and that the role of research should be to support that.

*2.5. Reflections on Moving beyond Race-Conscious Research and Scholarship*

As Gordon da Cruz and others have argued, in the policy and the legal arena, inequality persists because of color-blind approaches (Gordon da Cruz 2017; Bonilla-Silva 2006). As such, Gordon da Cruz (2017) argues that "[r]ace-conscious analyses can assist CES practitioners in avoiding research that could exacerbate existing inequities" (p. 374). We concur that researchers must constantly reflect on how their approaches contribute to the pathologizing of communities of color (Battle and Serrano, forthcoming). Furthermore, we find Gordon da Cruz's call for race-conscious approaches as tools to analyze the connections between legacies of institutionalized racism and inequality generative. However, regardless of how critical or race-conscious it is, research alone does not address inequality. In our CRRP approach, we find abolition imperatives and collective struggle to be a more useful tool to analyze the legacies of racial capitalism—including slavery, colonialism, the carceral state—and to guide us towards collective struggles and social transformation.

While notions of abolition continue to be routinely rejected in mainstream politics as hopelessly idealistic and impractical, the imminent need for abolition has always been clear to community-based organizers and grassroots activists in coalitions and organizations such as BSS. BSS constituents have suffered irreparable harm at the hands of policing and prison institutions throughout LA County. For example, in the Social Justice Learning Institute (SJLI), one of seven community-based organizations that make up BSS, Gabriel Regalado is charged with overseeing that their organizational praxis in youth justice advocacy maintains abolitionist imperatives. The application of CRRP is what primarily guides their approach.

SJLI not only holds youth and community narratives and strategic sensibilities at equal value to university research, but they also recognize the vitality of having the latter be guided by the former. SJLI rejects "grasstops" sensibilities in addressing the historic and extant injustices and inequities that plague our communities. In other words, their advocacy is not narrowly focused on stakeholders only, and they do not subscribe to vanguardianism in their organizational praxis. Their objective is to serve as a resource and platform for youth and community to engage in collaborative leadership development in resistance to issues of educational inequity, food and housing insecurity, and youth criminalization and carcerality. The intent is not to impose abolitionist politics on the community, but to serve as a vessel through which community-based abolitionist sensibilities can be further articulated and developed.

SJLI makes no tabula rasa assumptions about the political consciousness of youth and community members. They recognize that their lived experiences when navigating neoliberal realities are valid sources of knowledge that shape and inform their political consciousness. They also acknowledge that organic intellectual discourse—cultivated in barbershop discussions, music lyricism, and social media engagement—are also valid sources of knowledge. However, SJLI is also wary of how lived experiences and organic intellectualism can be vulnerable to co-optation and misdirection by the neoliberal project. Frantz Fanon ([1961] 2007) cautioned against the assumption that a sound comprehension of political and economic realities necessarily produces radical action. Therefore, our intent is to infuse the experiential knowledge and organic intellectualism in our communities with a transformative resistance (Solorzano and Bernal 2001) framework through critical pedagogy (Scorza et al. 2013) This is part of the CRRP approach at SJLI.

Since 2008, SJLI has implemented its Urban Scholars Program—an ethnic studies and leadership development class that engages Black and Brown high school students, particularly young men, in critical literacy and action research. Designed and developed by SJLI's founder and former Executive Director, D'Artagnan Scorza, the program aims to engage students in youth participatory action research with the aim of validating their experiential knowledge as legitimate basis for research. It seeks to embrace the potential in all students by offering them opportunities to name, explore, and analyze their experiences, and respect them as authors and experts of their own lives" (Mirra et al. 2016). Over the years, Urban Scholars has engaged in a myriad of youth-led research projects aimed at analyzing students' experiences with oppression, anti-Blackness, and xenophobia at their schools and harm in their neighborhoods. Some of BSS's youth leaders had their first exposure to CRRP approaches in the Urban Scholars program.

SJLI youth leaders, such as Emmanuel Karunwi, Amir Casimir, Sean Jones, and Amarion Charles have been intermittently involved in advancing BSS efforts like the Safety and Youth Justice Project, the Youth Bill of Rights, and the Police Free Schools campaigns in a multitude of ways—by offering public comment at Los Angeles Unified School District Board meetings, speaking at rallies and press conferences, consistently attending strategic meetings, and leading political education sessions—but they were also deeply involved in the creation of these campaigns. In other words, their intellectual and activist contributions helped build the abolitionist foundations embedded within these campaigns rather than their commentary being included as appendages to an existing critical framework. Herein we can identify the fundamental difference between community-*engaged* or community-*involved* scholarship and community-*rooted* participatory research—that the former incorporates community narratives as corroborating evidence to a university-driven study. The latter employs university resources and other critical theory devices to engage a community-driven study in which community members are involved in generating the research questions and shaping the research design.

## 3. Conclusions

In a conversation between Mariame Kaba and Eve Ewing, Kaba shared that her "unit of impact" as a prison abolitionist and organizer is developing and strengthening relationships (Kaba 2021). Many BSS youth leaders have been harmed or have witnessed how their loved ones have been harmed by policing, criminalization, deportation, and organized abandonment (Gilmore 2007). Thus, our CRRP approach is rooted in being in community, building relationships, collective struggle, and guided by abolitionist imperatives. We also want to acknowledge that our BSS Safety and Youth Justice Project is nested within a particular carceral history in Los Angeles (Hernández 2017; Felker-Kantor 2018; Sojoyner 2016). Nonetheless, we envision our CRRP approach as a recommendation and, most importantly, a reflective invitation for individuals and collectives conducting or considering community rooted projects. We also want to acknowledge that this approach might not be possible for all, not because their interest is lacking but because university structures and processes might infringe community rooted approaches to research.

Yet, we remain optimistic and offer our CRRP approach as a process and reflective tool to interrogate what it means to be *rooted* in community vs. partnered with community. Being rooted comes with a different praxis, and the challenge for us all is: how do we move forward foregrounding abolitionist praxis? With abolition, the Black Radical Tradition, youth organizing, and collective struggle as guiding frames for praxis, we reflect on questions about knowledge, expertise, racial analysis, and abolition with young people and their material conditions always in mind. Specifically, we highlight our CRRP approach by extending Gordon da Cruz's generative call for critical community engaged scholarship by offering our *community rooted and research praxis* approach. We offer our CRRP approach not as a totalizing model for community engaged research but as an invitation for community driven researchers to imagine what it would mean to occupy a "criminal relationship" with the university.

Our CRRP approach has meant using university resources, knowledge, and other tools to support youth as they developed the BSS Safety and Youth Justice Project. For us, taking a CRRP approach as university affiliated researchers has meant being invested in realizing youth's visions of social transformation. Particularly, their abolitionist calls to reimagine schooling, safety, and well-being in Los Angeles County and beyond. Thus, we offer BSS's CRRP approach as work-in progress because we recognize that collective struggle means constantly reorienting and building new currents of resistance and transformation, given the persistent nature of racial capitalism and the carceral state. Yet, it is in enduring abolitionist struggles and BSS youth leaders that we find tools to push the possibilities and limitations of community driven research.

For researchers—in and out of the university—conducting YPAR projects, we also offer our CRRP approach as a tool to avoid engaging in what Clay and Turner (2021) describe as "managerialist subterfuge". In a recent study of two YPAR projects, Clay and Turner (2021) found that adults often co-opted youth's activist agendas and intervened because youth were presumed to need of adult management and development. As such, for those considering our CRRP approach as a framework, we invite you to ask yourself the following questions:

> (1) What is it that young people envision for themselves and their communities that makes me uncomfortable? (2) How does that affect my interactions with them? What specific tools are young people missing that I can provide? (3) How can I help them understand the context and political landscape, while still empowering them to make decisions that best support their visions? (4) How can I encourage and amplify the voices of youth and community members in ways that go beyond typical state-sanctioned bureaucratic solutions (e.g., taskforces, study groups, work groups, etc.)? (5) In what ways can I return decision-making power to youth and community members over public processes and decisions that extend beyond traditional civic engagement approaches [and the universities professed commitment to "public good"]? (Clay and Turner 2021, p. 30).

With those questions in mind, we hope that our CRRP approach encourages readers to move toward abolitionist imperatives that examine and confront the systems and state actors that harm Black youth and youth of color.

**Author Contributions:** Conceptualization, U.S., D.C.T.III, G.R. and A.B.; formal analysis, U.S., D.C.T.III, G.R. and A.B.; resources, U.S., D.C.T.III, G.R. and A.B.; writing—original draft preparation, U.S., D.C.T.III, G.R. and A.B.; writing—review and editing, U.S., D.C.T.III, G.R. and A.B.; visualization, D.C.T.III. All authors have read and agreed to the published version of the manuscript.

**Funding:** This research received no external funding.

**Institutional Review Board Statement:** Not applicable.

**Informed Consent Statement:** Not applicable.

**Data Availability Statement:** Not applicable.

**Acknowledgments:** We would like to thank the reviewers for their feedback and editors, Claudia Lopez and Steven McKay. Most importantly, we would like to thank our partners in the movement and the Brothers, Sons, Selves Coalition specifically for all the work they do to transform the everyday conditions of young people.

**Conflicts of Interest:** The authors declare no conflict of interest.

## Notes

[1]　Bianca Baldridge (2019) defines youth workers as those who mentor, guide, and teach youth in various spaces, including grassroots youth groups, afterschool community-based programs, schools, and youth detention centers.

[2]　While others exist, here are a few examples of how youth organizations use research projects: VOYCE in Chicago, the Social Justice Learning Institute in Inglewood, Youth Justice Coalition in Los Angeles, and Black Youth Project.

[3]　Ignited by the police killings of George Floyd and Breonna Taylor amidst the onset of the COVID-19 pandemic.

[4]　The 8toAbolition framework was created by a collection of activists, writers, and scholars in response to liberal, harmful reformist policies that were proposed on the national level in response to the George Floyd/Breonna Taylor Summer uprisings of 2020. To learn more, please visit https://www.8toabolition.com/ (accessed on 9 February 2022).

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
