# Peer review of "Towards Community Rooted Research and Praxis: Reflections on the BSS Safety and Youth Justice Project"

_socsci, doi:10.3390/socsci11050195_

Round 1

Reviewer 1 Report

Thank you for the chance to read this article, and to learn about the critical work being done by youth and their adult supporters in LA. The example of BSS, and the CRRP framework, will be a valuable contribution to the published literature on participatory and community-based methods. It offers a strong critique of common approaches to research partnerships as well as a clear and compelling framework for scholars who want to push beyond partnership and collaboration toward collective struggle. The article is well written, passionate, and rooted in compelling examples of individual youth doing the work.

My thoughts on improving the article don't challenge the central argument, analysis, or structure of the paper. They have to do with 1) Details about what the research phases of CRRP looked like with BSS, and 2) the other writing that this work is in conversation with.

1. Research Details

While the article has good examples of youth using the findings of the BSS research, it glides very quickly over the process of developing the survey and analyzing the data. This, in my experience, is where a lot of the richness of YPAR is, where the rubber hits the road in terms of youth power and critical consciousness. The article could use a more expanded description of that process, either in the overview section (p. 6) or integrated into the later sections.

2. Using Gordon da Cruz's four questions really helped to structure the article and the four moves that the CRRP framework is making away from more typical engaged research models. However, I was not clear why it was chosen among the many frameworks out there. In fact, it seemed an odd fit after framing the project in terms of YPAR.

An explanation of why Gordon da Cruz was chosen would be helpful. Also, it will strengthen this framework to put it into at least a little bit of conversation with Critical Participatory Action Research (CPAR), which seems a closer cousin to CRRP than Critical Community Engaged Scholarship. I'm thinking in particular the work of Michelle Fine, Maria Elena Torre, and the Public Science Project https://publicscienceproject.org I think there is a lot more kinship there.

Also, I would love to see the authors add some references to other youth organizing initiatives that have used YPAR. Groups like VOYCE in Chicago. This would situate BSS within a wider tradition of YPAR led by youth organizers, rather than initiated by researchers, and point interested researchers toward side of PAR that doesn't get as many journal articles out there.

Other Points

It occurred to me that it might be helpful to have a two-column table outlining the differences between CCES and CRRP. I know that can be a bit simplifying, but the authors did a good job already of clearly naming the four big differences and that would be a way to help a reader visualize the overall framework. (for example, with rows like asset based vs. agency based, and partnering with community vs. in community) But I would leave that up to the authors.

I also note that the introduction names both abolitionist and Black radical traditions as influences on this framework. It seems like community/youth organizing is also a big influence, and might be mentioned up top to highlight that for readers.

Author Response

April 21, 2022

Dear Reviewers:

Re: Manuscript ID: socsci-1610576
Type of manuscript: Article
Title: Towards Community Rooted Research and Praxis: Reflections on the BSS
Safety and Youth Justice Project

Thank you for the invitation to resubmit this manuscript with edits. We are grateful for the constructive reviews and edits that have helped us strengthen this work. 

In the document attached, we use a table to outline and summarize how we have addressed the comments and issues identified by three reviewers. Please let me know if you need any additional information.  I look forward to hearing from you. And again, thank you for a thoughtful engagement with our work.

Sincerely,

Authors

Reviewer 2 Report

Thank you for the invitation to review the paper "Toward a community rooted research and praxis: reflections on the BSS safety and youth justice project ". As a community psychologist interested in YPAR, I found this paper to be thorough and engaging. The author(s) proposed a framework for Community Rooted Research and Praxis approach to extend Community Engaged Scholarship based off their reflections working with the Brothers, Sons, Selves (BSS) coalition. I appreciated the author(s) attention to the many theories within which YPAR is rooted, as well as a detailed description of the context. I have a few suggestions for improvement.

1. I recommend inclusion of a conceptual table or figure to illustrate the reflections presented in this paper. This could include visualizing the extensions of CES to CRRP, or perhaps focus only on CRRP and the reflection questions presented. 

2. A crystalized, succinct definition of CRRP and perhaps it's contrasts to other forms of participatory and/or CES would strengthen the paper. 

3. In the reflections provided for the four areas, were there any challenges or instances in the BSS project where CRRP did not play out, or perhaps "missed opportunities" for applying this approach? Might be useful to include throughout or in one paragraph in the conclusion.

4. In the conclusion, you reference useful questions from Clay and Turner (2021). What are some other specific recommendations for researchers who seek to utilize CRRP in their work.  Do you envision this as largely a reflective tool, or are there additional resources researchers could utilize to address each of these four areas?

Author Response

(The authors gave the same response as above.)

Reviewer 3 Report

The authors of this study have presented an intriguing and groundbreaking contribution to the scholarship on community rooted research and practice. It has been fascinating to read and I applaud the authors on this article but more importantly on their project and the succes they have had in the community. Below I am making some suggestions, which I hope will help the authors improve the manuscript.

  1. When reading the manuscript, I sometimes felt that readers who are not very familiar with the theoretical grounding of CRRP could benefit from a bit more explanation of concepts and movements to which the authors allude throughout the text as foundational to their argument. I suggest that clarifying the meaning of the terms will aid readers in understanding the connections of the author's CRRP to previous thought better. These concepts are (+lines):
  • radical capitalism (37)
  • CES (58-59)
  • Black Radical Tradition (60)
  • the definitions to the concepts named but not explained in lines 108-111
  • intersectionality (112) - much clearer later on but here would be good to know what it means when the term is first used
  • positionality (113)
  • critical race theory (131)
  • heteropatriarchy (172)
  • "grasstops" sensibilities (470)

2. During the beginning of the manuscript, I sometimes had the impression that my understanding of the author's argument would have benefitted if their arguments/illustrations of existing scholarship and their positioning toward them were better clarified through examples. Instances in which that would have been helpful are:

  • oppositional strategies (39) - may give some examples(?)
  • defunded (41) - can you give examples for the strategies used? and also explain what defunded means?
  • end of second paragraph of the introduction (after 44) - maybe add some examples what the partners actually did
  • challenges compounded by the realities of urban education (126-127) - which ones? Can you name a few and perhaps elaborate on the implications they have?

3. There were a few points in the manuscript that were unclear to me for different reasons:

  • 45: "referred to as community engaged scholarship" - by who/where - please add source
  • figure 1 - (1) the figure is illegible. If you choose to leave it in, please redo it so it is easier to read. (2) There is no engagement with the figure in the text. If you choose to leave it in, there should be an engagement with what readers can see to make clear how the figure helps advance/support your argument in the text. To be honest, I wonder whether you need it at all. You already do an amazing job in the text itself.
  • paragraph that starts in line 119: I think it would be good to make clear here that this is your contribution to the literature and be explicit about it. I get a bit lost as to who the credit belongs. There is a similar situation in the paragraph that starts in line 151
  • 179: on the notion of democracy: you mention the term here but it democratic engagement and what that looks like in the rest of the project is not really described in those terms but rather by saying doing research in community. I wonder if this term here is helpful to the project or actually hindering.
  • paragraph starting in line 341: was reflecting the methodologies for research also part of the process and engaging as to what extent they can be used for justice? If so, in any way, it may make sense to add that here.
  • 464: Regalado - who is that or who are they? I am lacking context here to be able to follow
  • when you are talking about the vignettes and Professor Jones, as for example in lines 187 and 376, I am completely lost as I do not get the reference or context. Could you make that clearer? Who is that, what is the vignettes she is talking about and how do the vignette relate to the overall project that you are citing?

4. There were a few paragraphs that felt a bit cumbersume in terms of transition. I suggest the author's go back and check where they may add better transitions in the following in-between spaces:

  • before section 1.1, line 74
  • after line 115
  • after line 150
  • after line 263
  • after line 293
  • after line 367

5. spelling errors

  • 186: diffrent; Crtical
  • 193: ,² (reverse)
  • footnote 3: 8 to Abolition - not the same spelling as in text "8toAbolition"
  • 270: extra space before "As Moten..."
  • 315 + 317: repetition of move beyond - maybe finde something else(?)
  • 374: extra space: "an   asset"
  • 408: exra space "power    or"
  • 409: extra space after "2021;    Kwon"
  • 422: missing subject: "that X can"
  • 436: spelling error? " 'Zae"
  • 454: plural s missing "tool"
  • 466: CRPR
  • 492: extra space after "Rights,  and"
  • 496: extra space after "activist       contributions"
  • 531: missing quotation marks after "managerial subterfuge

6. Bibliography

  • 559: place missing
  • 561: title in italics?
  • 563: place missing
  • 568: title in italics?
  • 570: place missing
  • 581: United States as a place? City instead?
  • 583: place missing
  • 585: title in italics?
  • 588-589: title in italics? place missing
  • 591: place missing
  • 594: UK is not a place; city instead
  • 596: place missing
  • 598: place missing
  • 602: place missing
  • 604: place missing
  • 607: place missing
  • 609: United States is not a place; city instead
  • 613: United States is not a place; city instead
  • 616: place missing
  • 617: journal title in italics
  • 620: title in italics?
  • 622: pages missing
  • 624 place missing
  • 627: publisher not in italics; place missing; check placement of chapter pages and stay consistent for all anthology chapters
  • 630: title in italics?
  • 627: publishing house should not be in italics; place missing
  • 630: United States is not a place; city instead
  • 635: unclear what source type this is - please check and correct
  • 639: access date missing
  • 642: add web address of report and access date
  • 644: title in italics; place missing
  • 647: journal title in italics
  • 650: access date missing
  • 652: United States is not a place; city instead
  • 666: title of anthology in italics; check placement of chapter pages and stay consistent; place missing
  • 668-669: check formatting of this chapter from an anthology: ch title should not be in italis, title of book in italics, eds should be named, add chapter pages, publisher and place
  • 672: journal title not in italics

Author Response

(The authors gave the same response as above.)
